


# Impact of Weather Patterns and Meteorological Factors on PM$_{2.5}$ and O$_3$ during the Covid-19 Lockdown in China

Fuzhen Shen[1*], Michaela I. Hegglin[1,2*], Yue Yuan[3]

[1]Institute of Energy and Climate Research, IEK-7: Stratosphere, Forschungszentrum Jülich, 52425 Jülich, Germany
[2]Department of Meteorology, University of Reading, Reading, RG6 6BX, UK
[3]Jining Meteorological Bureau, Shandong 272000, China

*Correspondence to*: Fuzhen Shen (f.shen@fz-juelich.de); Michaela I. Hegglin (m.i.hegglin@fz-juelich.de).

**Abstract.** The abnormal haze event in NCP (North China Plain) and the decline in ozone levels in SC (Southern China) from 21$^{st}$ January to 9$^{th}$ February 2020 have attracted public curiosity and scholarly attention during the COVID-19 lockdown. Most previous studies focused on the impact of atmospheric chemistry processes associated with anomalous weather elements in these cases, but fewer studies quantified the impact of various weather elements within the context of a specific weather pattern. To identify the weather patterns responsible for inducing this unexpected situation and to further quantify the importance of different meteorological factors during the haze event, two scenarios are employed. These scenarios compared observations to climatology averaged over the years 2015-2019 and the 'Business As Usual' (hereafter referred to as BAU) emission strength, using a novel structural SOM (Self-Organising Map) and ML (Machine Learning) models. The results reveal that the unexpected PM$_{2.5}$ pollution and O$_3$ decline from the climatology in NCP, North East China (NEC), and SC could be effectively explained by the presence of a double-centre high-pressure system. Moreover, the ML results provided a quantitative assessment of the importance of each meteorological factor in driving the predictions of PM$_{2.5}$ and O$_3$ under the specific weather system. These results indicate that temperature played the most crucial role in the haze event in NCP and NEC, as well as in the O$_3$ decline in SC. This valuable information will ultimately contribute to our ability to predict air pollution under future emission scenarios and changing weather patterns that may be influenced by climate change.

## 1 Introduction

The coronavirus disease 2019 (COVID-19) pandemic has lasted for three and half years and has led to over 6.9 million deaths globally as of June 2023 (Who, 2022). The Chinese government implemented strict lockdown measures nationwide during the first two months of 2020 to curb the spread of this pandemic (Le et al., 2020), which led to significant reductions in anthropogenic emissions, especially in the transportation sector (Xu et al., 2020; Wang et al., 2021; Liu et al., 2021). As a result, a decline not only in NO$_2$ but also in PM$_{2.5}$, PM$_{10}$, SO$_2$, and CO concentrations on a national scale was indicated by both satellite and ground-based measurements albeit with the negative consequence of enhancements in O$_3$ concentrations (Shen et al., 2022; Liu et al., 2021; He et al., 2020). Contrary to the situation in other regions from 21$^{st}$ January to 9$^{th}$ February 2020, Northern China (NC) and Southwestern China (SC) experienced severe haze pollution and decreased O$_3$ situations,



respectively (Le et al., 2020; Huang et al., 2021; Wang et al., 2020). This exceptional situation during the haze event in China thus lends itself to a large-scale 'experiment' to study the unusual phenomenon driven by atmospheric chemistry and meteorology.

$PM_{2.5}$ and ground-level ozone ($O_3$), especially in highly polluted regions, adversely affect human health (Lelieveld et al., 2015), agriculture (Feng et al., 2015; Wang et al., 2007), and the Earth's radiation budget (Liao et al., 2015; Dang and Liao, 2019) thereby leading to premature mortality, decreases in crop yields, and altering the climate. Anthropogenic $PM_{2.5}$, in addition to being generated by fossil fuels and biomass burning, is also produced through the reactions of inorganics (e.g. NO, $NO_2$, $SO_2$, $NH_3$, etc.) and Volatile-Organic Compounds (VOCs)(Zheng et al., 2017). In contrast, $O_3$ is not directly emitted but

is formed through a series of photochemical reactions involving multiple precursors (e.g., carbon monoxide (CO), methane ($CH_4$), VOCs, NO, $NO_2$, etc.)(Ge et al., 2013). Apart from intense local primary emissions and secondary chemical formation, stagnant meteorological conditions and regional transport are two additional contributors to severe haze and $O_3$ pollution events (Shen et al., 2020). Recently, a series of air quality regulations (Clean Air Plans, CAPs) released by the Chinese government have resulted in a notable decrease in anthropogenic emissions, leading to a substantial improvement in air quality due to

reductions in $PM_{2.5}$ concentrations, but a nationwide enhancement of $O_3$ pollution in China (Shen et al., 2020; Li et al., 2019b). It is known that the impacts of meteorological conditions and atmospheric chemical processes could result in non-linear responses of $PM_{2.5}$ and $O_3$ to the decreases in their precursor concentrations (Li et al., 2019a; Li et al., 2020). However, the specific responses of air pollutants and atmospheric chemistry to emissions and meteorological conditions have not been clearly determined.

For the haze event in China introduced above, recent studies on the topic suggested that complex atmospheric chemistry processes triggered by emission reductions and meteorological conditions are responsible for the unexpected haze formation and $O_3$ downward trend during the COVID-19 lockdown (Le et al., 2020; Fu et al., 2021). In detail, the substantial decrease in $NO_2$ emissions during the COVID-19 lockdown resulted in an increase in $O_3$ levels and nighttime $NO_3$ radical formation, enhancing the atmospheric oxidation capacity (AOC) and facilitating the formation of secondary aerosols. Additionally, the

presence of anomalous relative humidity promoted heterogeneous chemistry processes (Le et al., 2020; Huang et al., 2021; Ma et al., 2022). After the formation, more generated secondary aerosols were transported toward the in-situ measurement station in northern China (Lv et al., 2020). Meanwhile, some research pointed out that the high ambient humidity is also the key to the NC haze from the perspective of adjusting pH to control the formation efficiency of nitrate aerosol, which is one of the major species for NC haze (Chang et al., 2020; Sun et al., 2020). In addition to the influence of changes in chemical

reactions, a physical mechanism known as aerosol-planetary boundary layer (PBL) interaction is also considered to have had a significant impact on the haze formation (Su et al., 2020). For $O_3$, the decline in SC was attributable to the weakened photochemistry reactions due to the emission reductions in and the clean air masses' dilution effect on the mass loadings of NOx and VOC (Fu et al., 2021; Liu et al., 2021). Overall, meteorological conditions always played a critical role: High relative humidity is the trigger of aerosol heterogeneous chemistry by adjusting the particle pH or providing a reaction medium.

Meanwhile, the transport of the secondary aerosol or clean air masses and shallow PBL height are primarily driven by wind





and pressure, respectively. Importantly, the above weather elements are modulated synergistically by synoptic-scale weather patterns (SWPs) or large-scale atmospheric circulations.

Numerous studies have been conducted worldwide to explore the direct connections between SWPs and air quality Fields (Dayan and Levy, 2002; Demuzere et al., 2009; Pope et al., 2015; Hegarty et al., 2007; Bei et al., 2016; Jiang et al., 2017),

indicating that good air quality conditions are often observed under cyclonic weather systems, while poor air quality is frequently associated with anticyclonic conditions. However, the relationship between air quality and SWPs can differ depending on location, time and pollutants (Jiang et al., 2017; Liao et al., 2017). The methods for SWPs employed in these studies can generally be categorized into three groups: subjective (manual), mixed (hybrid), and objective (automated) (Huth et al., 2008). Objective classification methods for SWPs are known for their speed, objectivity, and high reproducibility, often

achieving 100%. On the other hand, manual approaches for SWPs have the advantage of allowing the user to control the selection of representative weather types (Lewis and Keim, 2015). Hybrid classification combines the strengths of both manual and automated techniques, where the users define the classification types, but the classification process itself is performed automatically (Frakes and Yarnal, 1997; Lewis and Keim, 2015; Huth et al., 2008). At present, the subjective method was used to investigate the contribution of six SWPs to $PM_{2.5}$ pollution in Northwest China (Bei et al., 2016). While subjective

approaches are suitable for analyzing short time series, they have significant limitations when applied to large datasets spanning extended periods of time (Chen et al., 2022). Hybrid classification for SWPs is more popular than the subjective one and was applied to explore the impact of SWPs on $O_3$, $PM_{2.5}$ and CO in NCP, Yangtze River Delta (YRD), and Eastern China, respectively (Zhang et al., 2013; Zhang et al., 2016; Han et al., 2018; Liao et al., 2017). As an objective classification and with its advantages, the self-organising map (SOM) algorithm has been used to identify the impact of different SWPs on $O_3$ and

$PM_{2.5}$ in YRD and Sichuan Basin (SCB), respectively (Shu et al., 2020; Zhan et al., 2019). In addition, the principle component analysis T-mode, k-mean clustering and other clustering approaches (like the Lamb-Jenkinson method) also were adopted to quantify the impact of SWPs on $O_3$ in NCP (Miao et al., 2017; Dong et al., 2020; Liu et al., 2019).

Based on the studies mentioned above, previous research on the drivers for unusual haze and $O_3$ decline events has concentrated on the influence of atmospheric chemistry processes accompanied by the anomalous of one or two weather

elements, but has not yet focused on the impact of weather elements in a comprehensive and synergistic way. Therefore, we here investigate the effect of anomalies in weather conditions with respect to climatology on $PM_{2.5}$ and $O_3$ concentrations during the haze event in the COVID-19 lockdown, specifically. To this end, we apply a novel SOM algorithm called structural SOM (S-SOM) to identify the most meaningful clustering number of weather patterns and compare it to other traditional SOM methods including ED-SOM and the SOM algorithm based on the Pearson correlation coefficient (hereafter named COR-

SOM). Furthermore, after determining the weather patterns, we evaluate the contribution of SWPs to $PM_{2.5}$ and $O_3$ changes during the COVID-19 lockdown in China. At last, to better understand what role each meteorological factor played in the $PM_{2.5}$ and $O_3$ pollution during this period, the SHapley Additive exPlanations (SHAP) approach is used to evaluate their relative importance for the predictions of the Machine Learning model. The knowledge gained ultimately will help to predict air pollution under future emission scenarios and weather patterns potentially altered by climate change.



## 2 Method

### 2.1 Observational and model dataset sources

The hourly observation dataset from 2015 to 2020, including two air pollutants ($PM_{2.5}$ and $O_3$) and six meteorological factors (Pressure: P, Precipitation: Precip, Temperature: Temp, Relative Humidity: RH, Wind Speed: WS, Wind Direction: WD), was divided into two parts: training dataset and test dataset, used to build a prediction model based on Machine Learning. Air pollutant and meteorological station datasets were downloaded from the National Environmental Monitoring Center (http://www.cnemc.cn) and the National Meteorological Science Data repository (https://data.cma.cn). To better understand the climatology impact on air pollutants, 367 surface measurement stations across China are divided into eight different regions (including NCP: North China Plain; IM: Inner Mongolia; NEC: North Eastern China; YRD: Yangtze River Delta; CS: Central South; SC: Southern Coast; TP: Tibet Plateau; NWC: North Western China) based on different typic climate characteristics (climate classification scheme link: https://www.resdc.cn/data.aspx?DATAID=243, **Fig. 1**). Hourly surface ERA5 data, including Mean Sea Level Pressure (MSLP) (at 14:00 local time per day), and total Solar Radiation (SR), were retrieved from the European Centre for Medium-Range Weather Forecasts (ECMWF).

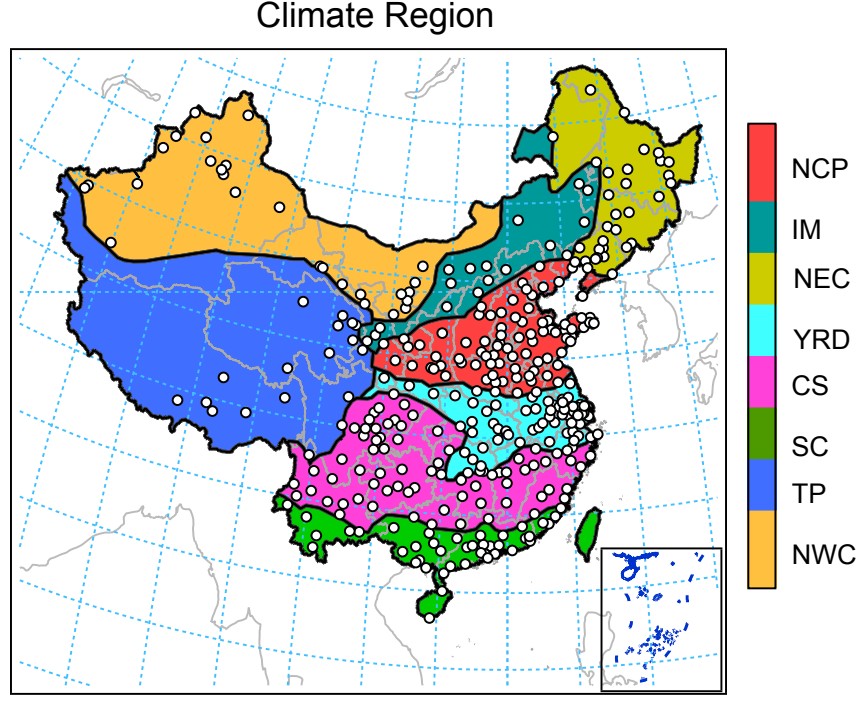

**Figure 1: The spatial distribution of air quality measurement stations in different climate regions (circles represent surface measurement stations, colors indicate different climate zones. NCP: North China Plain; IM: Inner Mongolia; NEC: North Eastern China; YRD: Yangtze River Delta; CS: Central South; SC: Southern Coast; TP: Tibet Plateau; NWC: North Western China)**



## 2.2 Structural SOM algorithm (S-SOM)

For the SOM algorithm, it involves iterative learning processes that progressively update the nodes in the output map until they converge to a stable solution. During each learning step, the SOM algorithm selects an input vector in a random way and then searches for a node that best matches that particular vector. Traditionally, the Euclidean Distance (ED) in the SOM algorithm is often used as a criterion to search for the winning node that is closest to an input vector. ED is very popular in the SOM algorithm but with significant shortcomings when applied to compare structured inputs with temporal or spatial orders. As a result, the limitations of ED become particularly significant in climatology research, where the data are often with a

spatial and temporal structure, which might result in the degradation of the spatial correlations between air pressure patterns in weather maps (Doan et al., 2021).

The S-SOM algorithm is executed following the procedure proposed by Kohoen and is widely used in many studies (Kohonen, 1982). To begin, an S-SOM is initialized by configuring the SOM node and determining the number of training iterations. The training process involves three key steps:

1. Selecting an input vector.

    2. Identifying the best matching unit in the SOM for the input vector.

    3. Updating the weight vectors of the SOM nodes using specific parameters.

The only difference between the traditional SOM and S-SOM is that the similarity index (S-SIM) rather than ED is used to compare the similarity between vectors. S-SOM was first proposed by Wang et al. (2004) and can be expressed in the following

equation.

$$\mathrm{S-SOM}(x,y) = \left[ l(x,y)^\alpha \times c(x,y)^\beta \times s(x,y)^\gamma \right] \qquad (1)$$

Here, $x, y$ are two vectors, and $l, c, s$ are three comparison measurements representing luminance, contrast and structure, respectively. The three comparison functions are as follows:

$$l(x,y) = \frac{2\mu_x\mu_y + c_1}{\mu_x^2 + \mu_y^2 + c_1} \qquad (2)$$

$$c(x,y) = \frac{2\sigma_x\sigma_y + c_2}{\sigma_x^2 + \sigma_y^2 + c_2} \qquad (3)$$

$$s(x,y) = \frac{\sigma_{xy} + c_3}{\sigma_x\sigma_y + c_3} \qquad (4)$$

Here, the average and standard deviation values are represented by $\mu, \sigma$, respectively. The parameters $c_1, c_2, c_3$ are used to stabilize division operations involving a weak denominator. The luminance, contrast, and structure in the S-SOM formula are three elements of human perception. Luminance assesses the similarity in brightness values between images. Contrast

quantifies the similarity in illumination variability among images. Lastly, the structure measures the correlation in spatial interdependencies between images, reflecting how the spatial elements of the images are related to each other (Wang and



Bovik, 2009). Here, we can set the values of c1, c2, and c3 to 0, and the weights α, β, and γ to 1 to simplify the model (Doan et al., 2021). The final expression shows as:

$$S - SOM(x, y) = \frac{(2\mu_x y)(\sigma_{xy})}{(\mu_x^2 + \mu_y^2)(\sigma_x^2 + \sigma_y^2)} \qquad (5)$$

As the function shows, the S-SOM ranges from -1 to 1. A value of 1 indicates complete similarity, while a value of -1 indicates complete dissimilarity. S-SOM offers robust, user-friendly, and comprehensible alternatives to the conventional ED approach, particularly when dealing with datasets with spatial and temporal order (Wang and Bovik, 2009).

### 2.3 Machine learning model

The impact of meteorological factors on the variation of air pollutant concentrations is typically determined via chemical
transport models. However, these model predictions are associated with substantial uncertainty since they rely on the correct quantification of changes in the emission inventory of each city under multi-faceted anthropogenic air pollution interventions (e.g., clean air plans, and COVID-19 lockdown measures). Here, a Gradient Boosting Machine (GBM) model was trained with observations of meteorological factors, with the GBM being able to capture the location-specific characteristics and thus suitable for the prediction of air pollutant concentrations attributable to the impact of meteorology in different cities across
China. Observations of meteorological factors, together with time variables from 2015 to 2019, are considered as the training dataset to predict the concentrations of $PM_{2.5}$ and $O_3$ in China. The meteorological factors are listed as follows: P, Precip, Temp, RH, WS, and WD. The time variables include Julian Day (JD), Day of Week (DOW), Holidays, and the Chinese New Year (CNY) days in each year. After selecting the best ML model under cross-validation, a ML experiment was designed to make a prediction of $PM_{2.5}$ and $O_3$ in the first two months of 2020.

### 2.4 Shapley Additive ExPlanation (SHAP) Method

Quantifying the importance of input features of the ML model is as vital as the overall accuracy of the prediction itself. However, interpreting the higher accuracy achieved by ensemble or ML models on certain datasets can be a challenging task. To deal with this contradiction between higher accuracy and non-interpretability, SHAP, a game theory approach, is applied to calculate the importance value for each specific independent feature. In brief, the importance value of each feature is
attributed to the difference in one prediction output with one feature versus the prediction output without this corresponding feature. Using this approach, the importance value for each independent variable can be calculated in each prediction model. For each predicted model with n variables in one sample $(x_l)$ and the predicted output $f(x_l)$, the equation of the prediction function is described as follows:

$$f(x_l) = E_0(f, x) + \sum_{m=1}^{n} E_m(f, x_l) \qquad (6)$$

where $x_l$ is the input with variable $m$ in the prediction model $f$ generating the SHAP value of $E_m(f, x_l)$. $E_0(f, x)$ is the expected value for the prediction model over the whole dataset.




## 3 Results

### 3.1 Spatial variations of air pollutant and meteorology in climatology

The spatial distribution of the fractional differences in air pollutant concentrations during the haze event from 21st Jan to 9th

Feb 2020, calculated between mean values during the event in 2020 and the values averaged over the same period from 2015 to 2019, for all six air pollutants are shown in **Fig. 2**. Half of the climate regions, including YRD, CS, SC, and TP, showed different magnitudes in decreases (increases) (**Table 1**) from the climatology for $PM_{2.5}$, $PM_{10}$, $NO_2$, $SO_2$, and CO ($O_3$), primarily attributed to the significant anthropogenic emission reduction during the COVID-19 lockdown (Nie et al., 2021; Wang et al., 2022; Shen et al., 2022). However, contrary to expectations, $PM_{2.5}$ concentrations did not drop as anticipated at

the beginning of the lockdown in NCP, IM, NEC and NWC. Instead, these regions experienced an unexpected increase of 8.6%, 31.8%, 22.3%, and 2% compared to climatology during the same period, respectively. In addition, $O_3$ showed an unexpected drop of -0.8% in SC when compared to climatology during the same period. Our recent work also found a -0.9% decline in $O_3$ compared to a climatology averaged over the years 2015-2019 during the COVID-19 lockdown across China (Shen et al., 2022). As a fractional difference from climatology, the spatial distribution of the key meteorological variables

RH, P, Precip, Temp, and WS are shown in **Fig. 3**. Generally, positive RH and negative WS anomalies are always accompanied by strong regional elevation of $PM_{2.5}$ in NCP, NEC, IM, and NWC. Positive P anomalies coupled with increased $PM_{2.5}$ demonstrate the most prominent regional characteristics in NEC. In SC, the most noticeable features were observed as a combination of hotspot Precip anomalies and decreased $O_3$ levels. Overall, the regional characteristics of $PM_{2.5}$ and $O_3$ all have a close relationship with different meteorological anomalies, which are usually controlled by the regionally prevailing SWP.

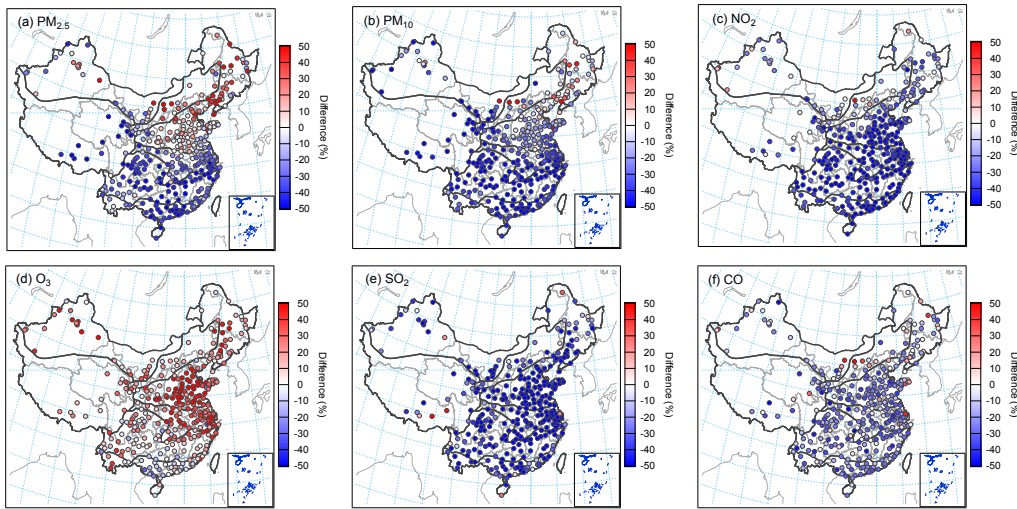


**Figure 2: The spatial distributions of fractional differences between mean values during the haze event in 2020 and the climatology over the same period of the years 2015-2019 for six air pollutants (including $PM_{2.5}$, $PM_{10}$, $NO_2$, $O_3$, $SO_2$, and CO).**





**Table 1. The climate characteristics and mean averages of six air pollutants in different regions across China.**

| Climate zone | Number of cities | Climate characteristics | $PM_{2.5}$ | $PM_{10}$ | $O_3$ | $NO_2$ | $SO_2$ | CO |
|---|---|---|---|---|---|---|---|---|
| NCP | 86 | Semi-humid warm temperate climate | 8.6% | -17.6% | 39.3% | -43.4% | -66.9% | -23% |
| IM | 17 | Semi-arid mid-temperate climate | 31.8% | -10.7% | 15% | -9.4% | -40.2% | -2.6% |
| NEC | 31 | Cold temperate climate, semi-humid, mid-temperate climate | 22.3% | 0.7% | 26.4% | -24.2% | -45.8% | -10.3% |
| YRD | 67 | Humid, north subtropical climate | -29.7% | -41.7% | 32.4% | -54.3% | -60.9% | -24.6% |
| CS | 68 | Humid, mid-subtropical climate | -41.4% | -50.9% | 11.5% | -53.2% | -56.7% | -25.2% |
| SC | 42 | South subtropical climate | -40.2% | -45.2% | -0.8% | -51.8% | -44.3% | -24.8% |
| TP | 9 | Plateau climate | -48% | -61.9% | 16% | -27.3% | -40.1% | -26.4% |
| NWC | 28 | Arid, mid-arid, mid-temperate climate | 2% | -34.1% | 28.7% | -14.5% | -49.6% | -12.3% |





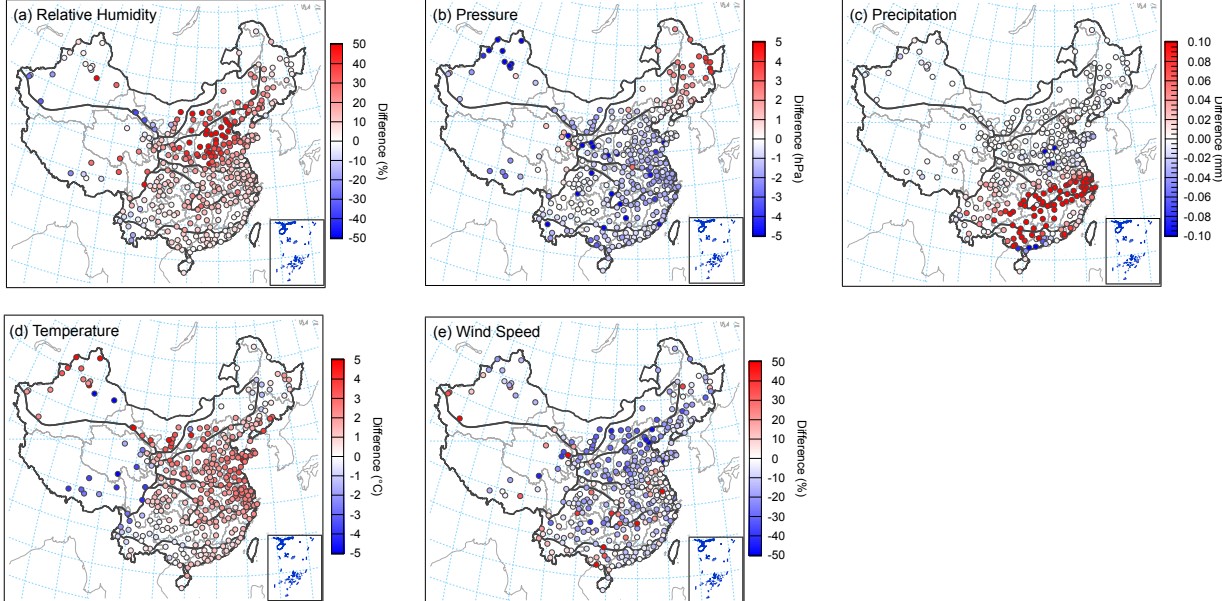

Figure 3: The spatial distributions of differences between mean values during the haze event in 2020 and the climatology over the
same period of the years 2015-2019 for meteorological factors (including Relative Humidity, Pressure, Precipitation, Temperature,
and Wind Speed).

## 3.2 Identification of the SWP during the unexpected haze event

To identify which SWP can regionally induce an unexpected $PM_{2.5}$ increase and $O_3$ reduction compared to climatology, three
different SOM methods were employed to identify different types of SWPs (from 2 to 8) by using MSLP data in the first two
months from 2015 to 2020 over China. **Fig. 4** shows MSLP patterns identified by S-SOM, COR-SOM, and ED-SOM running
three nodes, respectively. Taking this three-node analysis as an example, we can find that the three SWPs identified by S-SOM
(**Figs. 4a, 4b, and 4c**) are clearly distinct from each other. On the other hand, ED-SOM (**Figs. 4d and 4e**) and COR-SOM
(**Figs. 4g and 4h**) both classify two similar SWPs characterised by high-pressure systems over Siberia, thus resulting in a
failure of clustering. This interpretation is supported by the result of clustering number distributions for the three-node SWPs
(**Fig. 5d**). It should be noted that cluster numbers do not necessarily correspond to the same pattern between S- / ED-/ COR-
SOM. Here, it is found that S-SOM results are in a more 'ordered' clustering of nodes, where a prominent node (62.9%) is
accompanied by two non-dominant nodes (7.5% and 29.5%). On the other hand, both ED-SOM and COR-SOM exhibit
relatively similar cluster sizes with a percentage of 27%, 35.1%, and 37.9% for ED-SOM and 40.7%, 34.6%, and 24.7% for
COR-SOM, highlighting the prevalence of a more 'flat' clustering pattern. It can be concluded that the better classification
method for three-node SWPs is S-SOM with an 'ordered' clustering number distribution accompanied by a prominent node
(Doan et al., 2021). This consistent finding is also observed in other cases (**e.g. node numbers smaller or greater than 3,**



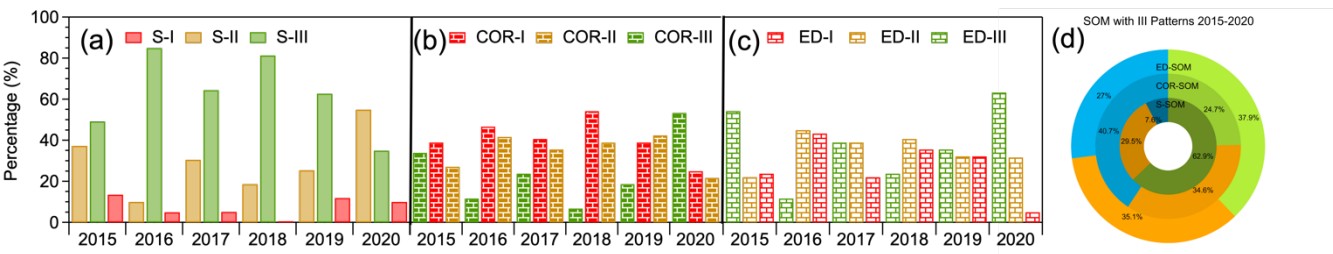

**Figure 4: Spatial distributions of three weather patterns for MSLP (Mean Sea Level Pressure) identified by S-SOM (a, b and c), COR-SOM (d, e, and f), and ED-SOM (g, h, and i) during the first two months from 2015 to 2020.**


**Figure 5: Cluster size distributions identified by S-SOM (inner ring), COR-SOM (middle ring) and ED-SOM (outer ring) over the years 2015-2020 (d) and days in each year (a: S-SOM; b: COR-SOM; c: ED-SOM).**

**Fig. S1-S12**). Then, we make a further comparison of the node number distribution of S-SOM (**Fig. 5a**), ED-SOM (**Fig. 5b**), and COR-SOM (**Fig. 5c**) in each year and find that S-SOM always has a prominent node with a value of more than 50% (2015:50%, 2016: 85%, 2017: 64%, 2018: 81%, 2019: 63%, and 2020: 55%) and the cluster sizes for ED-SOM and COR-



SOM are close to each other as well, which is consistent with a recent study indicating a better performance of S-SOM (Doan et al., 2021). Therefore, in addition to the algorithmic advantages, the characteristics of 'ordered' clustering nodes reinforce

the superiority of the S-SOM approach.

In terms of structure characteristics of clustering number distribution for S-SOM, three-node SWPs (**Fig. 4**) and seven-node SWPs (**Fig. S13**) were regarded as the optimal numbers of SWPs after checking the clustering number distribution for each run. From the top panel of **Fig. 4**, three types of SWPs identified by S-SOM demonstrate that NCP, YRD, NEC and NWC are under the control or influence of different high-pressure systems. For seven-node SWPs identified by S-SOM, even though

the high-pressure system varies in numbers and locations, some patterns (**Fig. S13d and Fig. S13e**) still have a relatively high similarity, which might be attributed to the over-splitting or a too short dataset to capture the full climatology. Overall, the result of three-node SWPs of S-SOM is thus identified as the best solution to study the haze event in China in further detail.

**3.3 Impact of weather elements on PM2.5 and O3 under the SWP**

To better understand the regional influence of different SWPs on PM$_{2.5}$ and O$_3$ concentration levels, NCP, NEC, and SC, which

have higher/lower than expected concentrations for PM$_{2.5}$/O$_3$ and have more measurement stations as well, were selected as the research domains, respectively. To investigate the cause of the unexpected PM$_{2.5}$ and O$_3$ variations with respect to climatology, a comparison of identified three-node SWPs is made between the days of 2020 and 2015-2019. As is shown in **Fig. 6** and as detailed in **Figs. 7-9**, pattern-I in 2020 (**Fig. 6d**) shows a North Coastal high-pressure circulation system, located in the Yellow Sea, which is enhanced from that in 2015-2019 (**Fig. 6a**) and influences the NCP and NEC regions (see **Figs.**

**7a,d-f and 8a,d-f**) more strongly from the southeast direction with a generally warmer and, in the case of NEC, also faster airflow. The double-centre high-pressure system in pattern-II is strengthened in 2020 (**Fig. 6e**) and located in the region of Mongolia and the Bohai Sea in China compared to 2015-2019 (**Fig. 6b**). This brings along a more stagnant, that is low-speed and cold, but in addition extremely wet, northern airflow controlling the NCP region (**Figs. 7a,d-f**) and a moderately wetter airflow dominating the NEC region (**Figs. 8a,d-f**). Pattern-III, on the other hand, shows a much weakened Siberian high and

a missing China north coastal high in 2020 (**Fig. 6f**), when compared to a pattern exhibiting two high-pressure centres during the 2015-2019 reference period (**Fig. 6c**). This leads to a generally warmer, slightly faster, and more humid airflow to the NCP (**Figs. 7a,d-f**) and NEC (**Figs. 8a,d-f**) regions. For SC, which is always located at the most southern part of the observed high-pressure centres (**Fig. 6**), and for all three patterns, only small changes are seen in 2020 compared to the 2015-2019 time period with a more easterly component in winds (**Figs. 9a-c**), leading to slightly warmer and (except for pattern III) moister airflow

(**Figs. 9 d-f**).





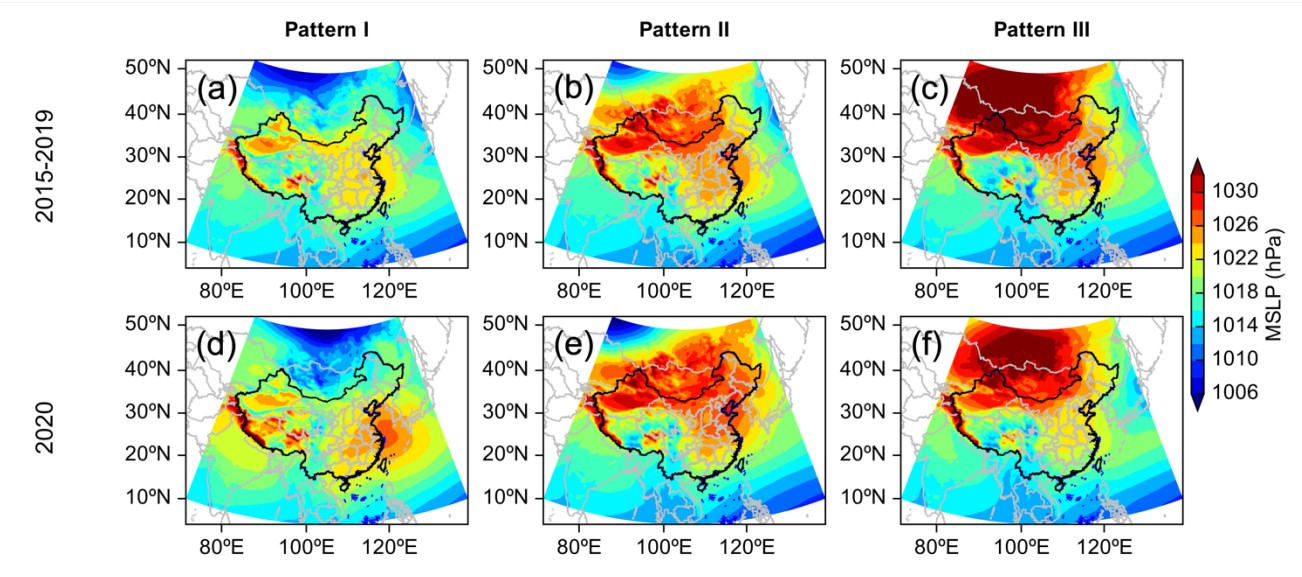

**Figure 6: Comparison of the three weather patterns between days in 2020 (d, e, and f) and 2015-2019 (a, b, and c), respectively.**


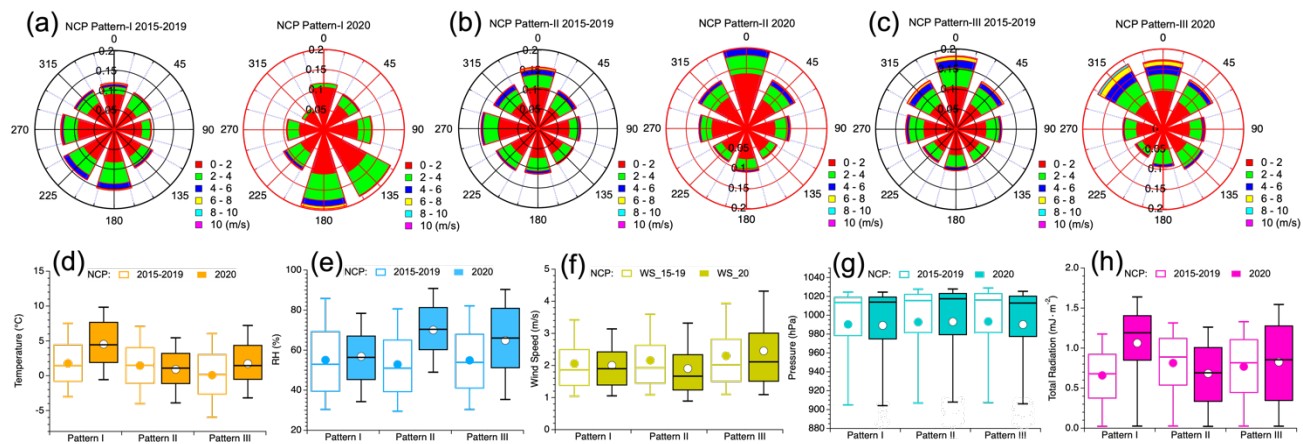

**Figure 7: Comparisons of different weather factors (including Wind Speed, Wind Direction, Temperature, Relative Humidity, Pressure, and Total Radiation) between days in 2020 (red rings and solid whisker-boxes) and in 2015-2019 (black rings and hollow whisker-boxes) for the three weather patterns in NCP.**





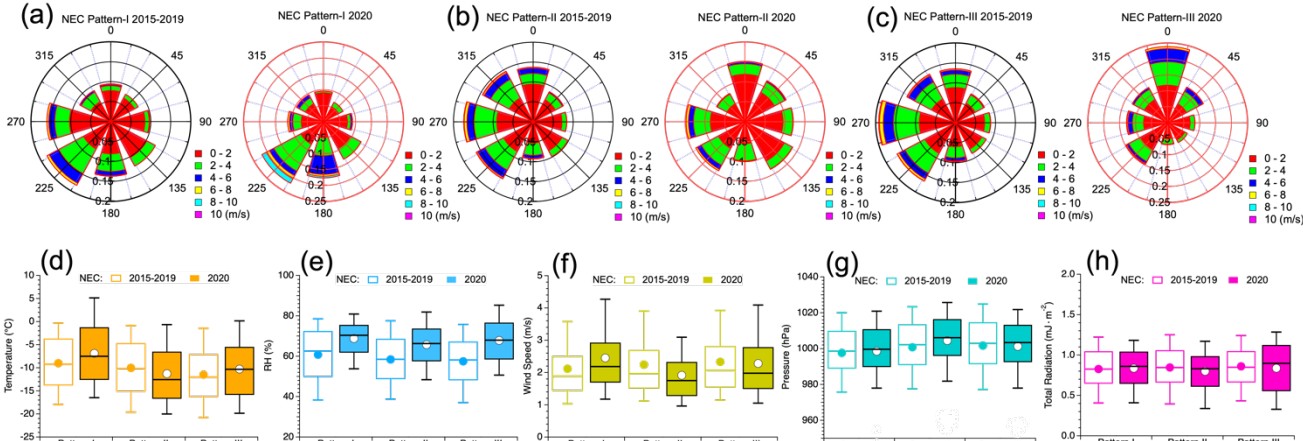

**Figure 8: Comparisons of different weather factors (including Wind Speed, Wind Direction, Temperature, Relative Humidity, Pressure, and Total Radiation) between days in 2020 (red rings and solid whisker-boxes) and in 2015-2019 (black rings and hollow whisker-boxes) for the three weather patterns in NEC.**

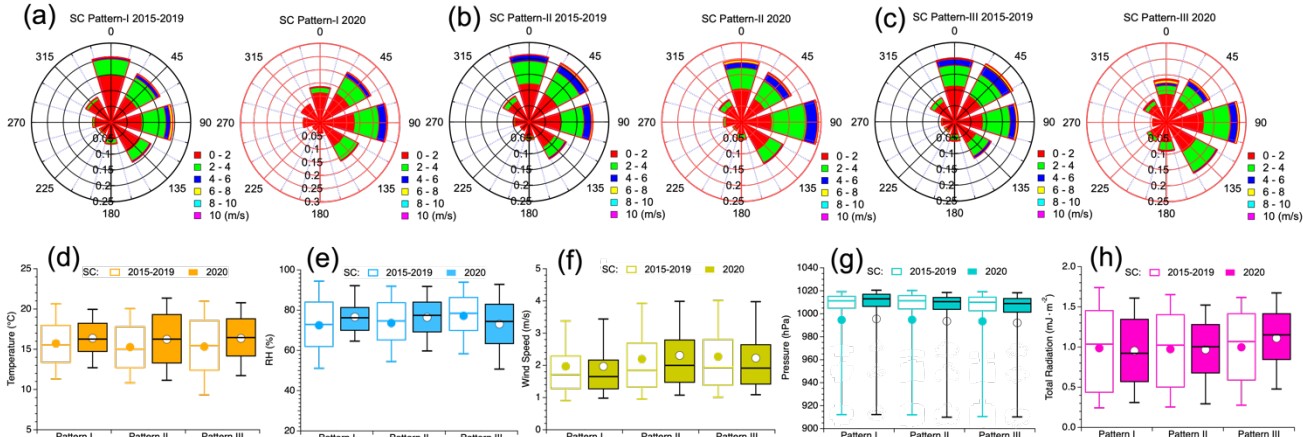

**Figure 9: Comparisons of different weather factors (including Wind Speed, Wind Direction, Temperature, Relative Humidity, Pressure, and Total Radiation) between days in 2020 (red rings and solid whisker-boxes) and in 2015-2019 (black rings and hollow whisker-boxes) for the three weather patterns in SC.**

We now turn to the discussion of the observed distributions of PM$_{2.5}$ and O$_3$ (**Fig. 10**) aggregated over the three SWPs and the regions NCP, NEC, and SC for the 2020 and the 2015-2019 time periods, respectively. For PM$_{2.5}$ in NCP (**Fig. 10a**), the mean values in pattern-I, II, and III in 2015-2019 all remained at high pollution levels with a value of 96.4, 92.6, and 87.7 µg/m$^3$, respectively. In contrast, due to the anthropogenic emissions reductions during the lockdown period in 2020, the PM$_{2.5}$ mean values for patterns I and III decreased to 68.8 and 59.8 µg/m$^3$ even coupled with a positive RH climatological anomaly (**Fig. 7e**: 2% and 10%), which could be conducive to generating additional PM$_{2.5}$ generally. Unlike pattern-I and III, the PM$_{2.5}$ mean value in pattern-II 2020 surprisingly keeps at an equivalent level (92.5 µg/m$^3$) to pattern-II in 2015-2019 (92.6 µg/m$^3$) under a weather condition of a combination of the greatest RH anomaly (**Fig. 7e**: 17%) and a negative WS anomaly (**Fig. 7f**:





-0.3 m/s), which offsets the contribution from the emissions reduction in NCP. For $O_3$ in NCP (**Fig. 10d**), the pattern-I and -

III in 2020 exhibit greater temperature anomalies (**Fig. 7d**: 2.7 °C and 2.9 °C; consistent with higher total radiation levels, see

**Fig. 7h**) and thus facilitate additional $O_3$ generation (20 µg/m$^3$ and 13 µg/m$^3$). The pattern-II in 2020 with a negative

temperature anomaly (-0.1 °C; consistent with lower total radiation levels, see **Fig. 7h**) is leading to a more moderate $O_3$

increase (3 µg/m$^3$).

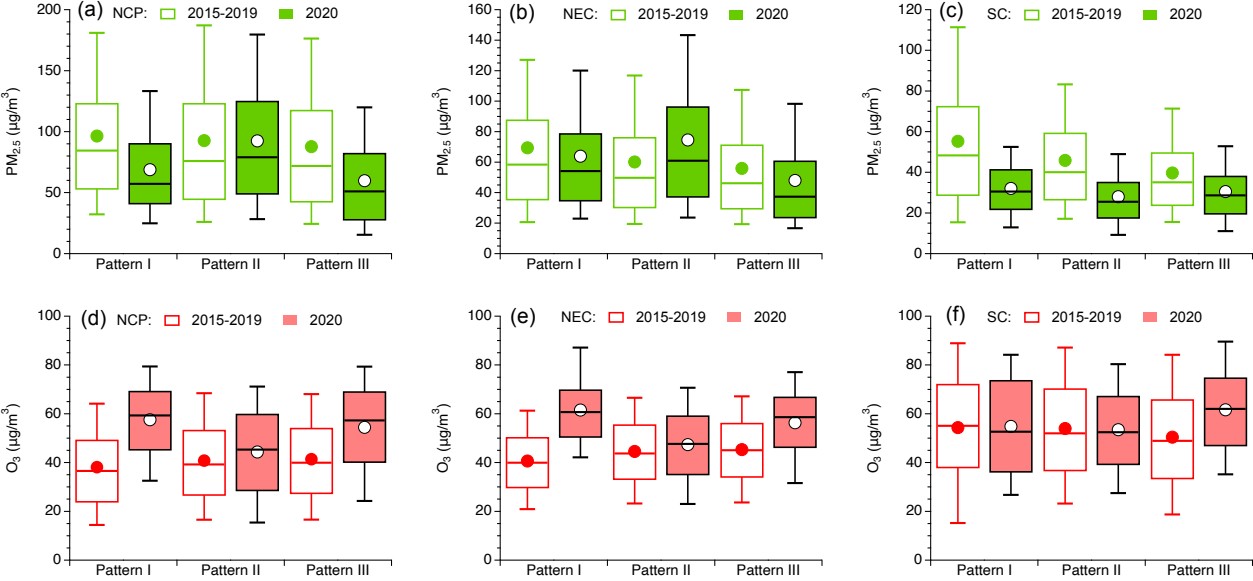

**Figure 10: Comparisons of PM$_{2.5}$ (green colour) and O$_3$ (red colour) between days in 2020 (filled whisker-boxes) and in 2015-2019 (hollow whisker-boxes) for the three weather patterns in NCP (a and d), NEC (b and e), and SC (c and f).**

In the NEC region, the maximum PM$_{2.5}$ increase (15 µg/m$^3$) occurred under the influence of pattern-II in 2020(**Fig. 10b**),

with a negative wind speed anomaly (**Fig. 8f**: -0.3 m/s) when compared to the same pattern in 2015-2019, indicating the

meteorological effect acts in the opposite way to the emission reductions during the COVID-19 lockdown period. Whilst,

without an offset effect from the unfavourable meteorological conditions, mean values of PM$_{2.5}$ for pattern-I and III in 2020

decreased by 5 µg/m$^3$ and 8 µg/m$^3$, respectively. For $O_3$ (**Fig. 10e**), unlike a negative temperature anomaly (**Fig. 8d**: -1.3 °C)

in SWP-II, both higher $O_3$ increases in SWP-I (11 µg/m$^3$) and III (11 µg/m$^3$) than that in SWP-II (2 µg/m$^3$) are driven by

positive temperature anomalies (**Fig. 8d**: 2 °C and 4.4 °C).

In the SC region, without an extreme weather element anomaly facilitating additional PM$_{2.5}$ production, PM$_{2.5}$ mean values

for all three SWPs in 2020 are at a lower level than in 2015-2019 (**Fig. 10c**), attributable to the emissions reductions during

the COVID-19 lockdown. Higher precipitation levels in 2020 than during the 2015-2019 period also helped reduce PM$_{2.5}$

levels (see **Figs. 3c and S14**). For $O_3$ (**Fig. 10f**), a negative RH anomaly (**Fig. 9e**) for SWP-III in 2020 has led to the greatest

$O_3$ elevation for this region. On the other hand, $O_3$ in pattern-I is found to remain at similar levels during both time periods

since no significant differences in weather patterns are found. Finally, a positive wind speed anomaly (**Fig. 9f**: 0.21m/s) is



conducive to an unusual O$_3$ decline (-0.5 µg/m$^3$) in SWP-II in 2020 when compared to 2015-2019, which is contrary to the O$_3$ situation under the effect of all other SWPs discussed above.

Overall, we found that the unexpected PM$_{2.5}$ pollution increase in NCP and NEC and an O$_3$ decline in SC occur simultaneously only during SWP-II, which is equivalent to the situation found in the observations during the haze event. When
we further investigate the calendar occurrences of the three different SWPs (**Fig. S15**), it is indeed found that 70% of haze days were associated with SWP-II. This finding thus indicates that SWP-II can be regarded as the representative weather pattern which best explains the cause of the unexpected haze and O$_3$ decline events.

### 3.4 Predominant meteorological factors for PM$_{2.5}$ and O$_3$ pollution

After identifying which SWP could control the impact of each weather element on PM$_{2.5}$ and O$_3$ levels as observed during the
haze event in 2020, we further use machine learning coupled with the SHAP approach to quantify the impact of each weather element on the PM$_{2.5}$ and O$_3$ under 'Business As Usual' (hereafter referred to as BAU) emission strength scenario during the haze event in 2020. This BAU scenario thereby represents a counterfactual to the situation under the Covid-19 lockdown which led to significant emission reductions. In our previous study, the GBM model was applied to train daily data over 2015-2019 and predict six air pollutants including PM$_{2.5}$ and O$_3$ over the first three months of 2020 in 367 cities across China (Shen et al.,
2022). The good performance of the GBM model was measured by achieving relatively high Pearson Correlation Coefficients (PCC) and lower root-mean-squared errors (RMSE) for the final predictions of PM$_{2.5}$ and O$_3$ (the details can be found in the supplementary materials). **Fig. 11 (a), (d), and (g)** and **Fig. S16** show the time series results in the first two months for PM$_{2.5}$ and O$_3$ between the observation and prediction in NCP, NEC, and SC respectively. We find that the predictions agree generally well with the observations with reasonably high PCCs (NCP: 0.7, NEC: 0.6, SC: 0.8), indicating the good performance of the
GBM model. Note that these predictions might be with high Root Mean Square Errors (RMSEs) due to the input being the BAU emissions instead of the lockdown emission reduction. In a second step, the SHAP module coupled to the GBM model was run to quantify the importance of the input variables during the haze event in 2020 (**Fig. 11 (b), (e), and (h)**). On average, in the BAU emission strength scenario, the SHAP value of time variables, including CNY, DOW, Holiday, and Julian Day, have no or negative impacts on PM$_{2.5}$ and O$_3$ (**Fig.11(c), (f), and (i)**). For meteorological elements that enhanced the production
of PM$_{2.5}$, temperature ranked first among the six meteorological elements during the haze event, followed by RH and pressure in NCP, versus RH and WS in NEC respectively. In contrast, RH in SC is the primary meteorological variable that facilitated the generation of O$_3$. Meanwhile, temperature and pressure play the opposite role to RH, leading to a reduction of O$_3$ surprisingly. When we investigate the observed weather elements in 2020 against that averaged over 2015-2019, we can find that NCP and NEC were both under the control of SWP-II with lower temperatures and a higher RH, which facilitate the
formation of PM$_{2.5}$. Meanwhile, the SC region was influenced by the SWP-II with higher temperatures, higher RH and higher WS weather conditions with the latter being favourable to the transport of air masses, resulting in a decline of O$_3$. Overall, we could not only find that the impact of weather elements on PM$_{2.5}$ and O$_3$ in the prediction scenario is consistent with that in climatology, but also can conclude that temperature plays a key role in such an impact.




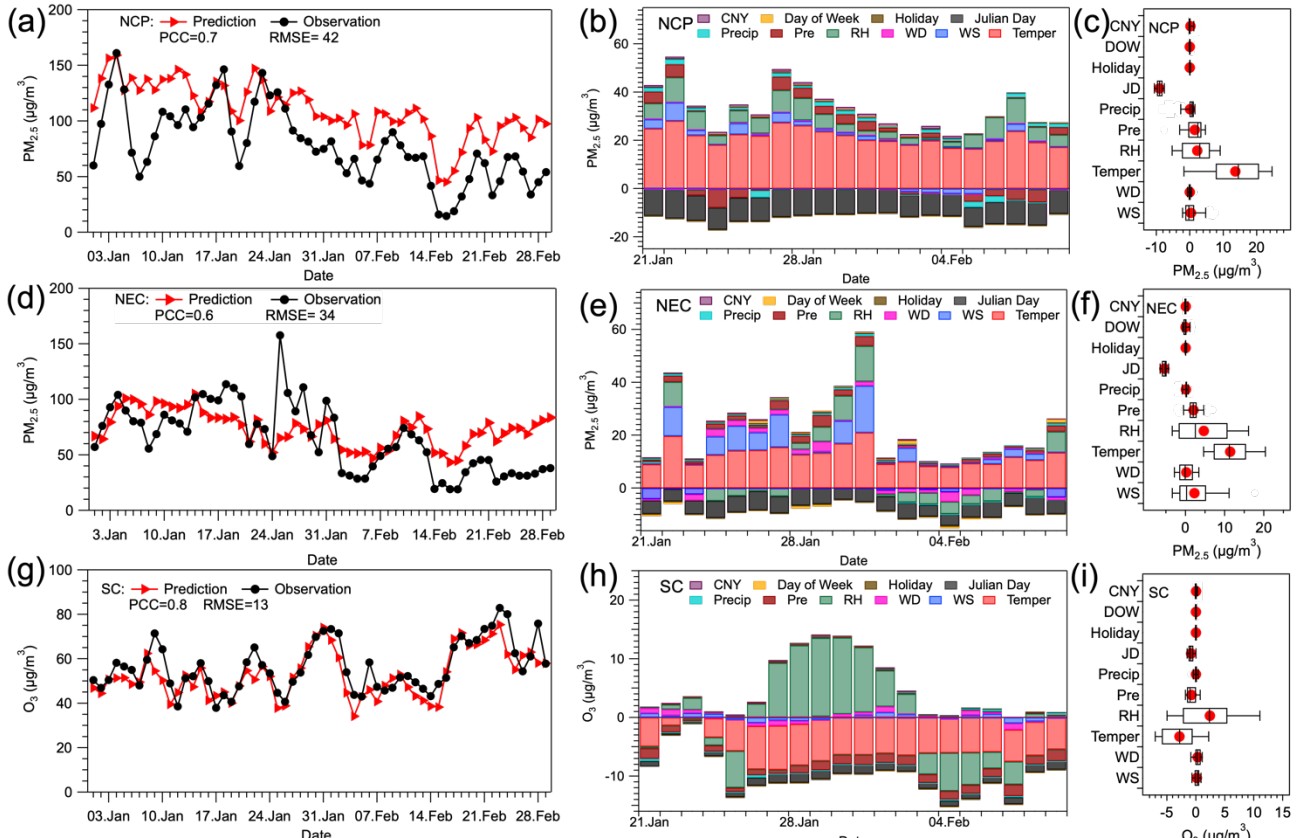

**Figure 11: Time series comparisons between observations (black dot line) and predictions (red triangle line) combined with the contributions from the input variables (colourful bar) to the PM$_{2.5}$ and O$_3$ changes in NCP (a and b), NEC (d and e), and SC (g and h) respectively. Note that the whisker-box plots represent the mean importance of the input variables during the prediction in NCP (c), NEC (f), and SC (i) respectively**

## 4 Conclusion

At the beginning of the COVID-19 pandemic, China suspended almost all non-essential human activities. However, serious haze pollution still occurred in North China during this period, triggering extensive investigations. On the other hand, whilst O$_3$ concentrations were increasing across almost all of China due to the shift in the chemical regime, the SC region exhibited a decrease in O$_3$. To further understand the role of meteorology in regulating air pollution during this period, we investigated in more detail the role of synoptic-scale weather patterns in driving the meteorology in these regions of China. To this end, we first determined the optimal approach for identifying synoptic-scale weather patterns out of three self-organising map methods. With the S-SOM method yielding the most optimal results, we then analysed the variation of each meteorological factor under the control of the weather type that produces anomalous PM$_{2.5}$ concentrations in the NCP and NEC, and anomalous O$_3$



concentrations in SC, and finally quantify the importance of each meteorological factor assuming a BAU scenario through a machine-learning model coupled with a SHAP module.

The large-scale double centre high-pressure system was identified by the optimal S-SOM method, which is with low-speed-cold-extremely wet-northern airflow controlling the NCP region, with low-speed-warm-wet airflow from the Bohai Sea dominating the NEC region, and with warmer air masses covering the SC region simultaneously. Whilst, the above weather elements anomalies controlled by the large-scale high pressure could well explain the unexpected $PM_{2.5}$ pollution and $O_3$ decline in climatology in NCP, NEC, and SC respectively.

Moreover, the SHAP results indicate that in the BAU scenario, the time series trend of $PM_{2.5}$ and $O_3$ have a high similarity with that of observations, indicating a good performance of the prediction model (despite the differing emissions). The SHAP results stress the impact of meteorological conditions on $PM_{2.5}$ and $O_3$ and further quantify the importance of each weather element under the specific weather system, revealing the most important role that temperature played in $PM_{2.5}$ pollution in NCP and NEC, and in $O_3$ decline in SC, respectively.

Overall, this study provides a potential way to understand the synergistic effects of various meteorological factors in reducing pollution and to quantify the importance of each weather element as well. As a result, the provision of information on what role each weather element plays in unexpected air pollution cases can help policymakers to implement air pollution control strategies. However, our work will have to be expanded further and add more related meteorological factors to our model to improve its performance. In fact, more studies should focus on the topic of understanding the impact of meteorology on

different air pollutants in particular due to weather conditions in a changing climate.

**Code availability.** Code is available upon reasonable request to the corresponding author (f.shen@fz-juelich.de).

**Data availability.** Data is available upon request to the corresponding author (f.shen@fz-juelich.de).


**Supplement.** The supplement related to this article is available online at:

**Author contributions.** FZS and MIH designed the experiments, FZS and YY conducted the numerical experiment and wrote the article. MIH supervise the idea and the experiment design. FZS, MIH, and YY discussed the experimental design and

analysis.

**Competing interests.** The contact author has declared that none of the authors has any competing interests.

**Disclaimer.** Publisher's note: Copernicus Publications remains neutral with regard to jurisdictional claims in published maps

and institutional affiliations.



**Acknowledgements.** We thank Yue Yuan from the Jining Meteorological Bureau for providing the meteorological dataset from observational station measurements.

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
