# Peer review of "Impact of Weather Patterns and Meteorological Factors on PM2.5 and O3 Responses to the Covid-19 Lockdown in China"

_EGUsphere, 2023_

## Author Comment (AC1)

Reply Letter to the Comments

**Manuscript ID:** Preprint egusphere-2023-2425

**Manuscript Title:** Impact of Weather Patterns and Meteorological Factors on PM$_{2.5}$ and O$_3$ during the Covid-19 Lockdown in China

**Authors:** Fuzhen Shen, Michaela I. Hegglin, Yue Yuan

Dear Editor, dear Reviewers,

We would like to thank you and the reviewers for the thoughtful comments regarding our manuscript. These comments have helped us greatly improve the interpretation of our findings, and are important for our future work as well. We have carefully revised the manuscript accordingly. Our point-to-point responses to the reviewers' comments, which are repeated in italics, are given below.

**To Reviewer #2**

*General comment:*

*This manuscript classified different types of synoptic-scale weather patterns during the first two months from 2015 to 2020 over China. Based on the ML (Machine Learning) models, the authors provided a quantitative assessment of meteorological factors in driving the predictions of PM2.5 and O3 under the specific weather system. The authors provided useful information about the anomalies of PM2.5 and O3 during the study period. This study is well within the scope of ACP. However, the discussions of relative results from the ML analyses were not well be demonstrated. While there is a need for minor revision, particularly in the discussion sections. I suggest that this paper could be published in the journal of ACP in case of the comments is addressed by the authors.*

**Authors' Reply #0:** We thank the reviewer for his/her overall positive assessment of our study. We have made modifications to our manuscript according to the reviewer's helpful suggestions below, which helped to improve the manuscript further.

*Highlight the new findings of this study. The authors should demonstrate the creative results, especially to differentiate those in previous studies. I think, studies on meteorological effects on driving the predictions of PM2.5 and O3 have been widely obtained. The authors should introduce more studies about them, and their comparisons with each other should be summarized and discussed in the "Discussion" part.*

**Authors' Reply #1:** We thank the reviewer for this comment. Indeed, many studies have estimated PM$_{2.5}$ and O$_3$ by using different prediction models[1], but they are limited to explain the final predictions[2-5], especially to provide details of specific input features. In our study, the SHAP module coupled to the GBM

model was run to quantify the local importance of the specific input variables during the haze event in 2020 (***Revised Manuscript version Line****:350-352*).

*References:*

[1]  Wu Y, Lin S, Shi K, Ye Z, Fang Y. Seasonal prediction of daily PM2. 5 concentrations with interpretable machine learning: a case study of Beijing, China. Environmental Science and Pollution Research. 2022 Jun;29(30):45821-36.

[2] Xiao Q, Chang HH, Geng G, Liu Y. An ensemble machine-learning model to predict historical PM2. 5 concentrations in China from satellite data. Environmental science & technology. 2018 Oct 24;52(22):13260-9.

[3] Zhang M, Wu D, Xue R. Hourly prediction of PM 2.5 concentration in Beijing based on Bi-LSTM neural network. Multimedia Tools and Applications. 2021 Jul;80:24455-68.

[4] Jin H, Chen X, Zhong R, Liu M. Influence and prediction of PM2. 5 through multiple environmental variables in China. Science of The Total Environment. 2022 Nov 25;849:157910.

[5]  Weng, X., Forster, G. L., and Nowack, P.: A machine learning approach to quantify meteorological drivers of ozone pollution in China from 2015 to 2019, Atmos. Chem. Phys., 22, 8385–8402, https://doi.org/10.5194/acp-22-8385-2022, 2022.

*Analytical method appeared adequate; however some key procedural and QA/QC details are missing. Please provide more details in the manuscript, including the time resolutions of field and reanalysis data, and the uncertainties of ML analysis.*

**Authors' Reply #2:** We thank the reviewer for his/her thoughtful point. More detailed descriptions of the data and ML method have been added to provide the missing information (***Revised Manuscript version Line****:113,405-407* ).

*Gradient Boosting Machine (GBM) was selected for a quantitative assessment of meteorological factors in driving the predictions of PM5 and O3. Does the authors try to compare it with other models, such as random forest, etc.*

**Authors' Reply #3:** We thank the reviewer for his/her comment. In fact, we had first used a flexible neural tree which is capable to perform automatic feature selection and function approximation[1]. However, the prediction results were not satisfactory and we decided to adopt the GBM model instead. The GBM model has some advantages for the prediction task, like better accuracy, higher efficiency, and capability of handling large-scale data[2]. As a result, GBM was implemented.

*References:*

[1]  Yuehui Chen, Bo Yang, Jiwen Dong, Ajith Abraham, Time-series forecasting using flexible neural tree model, Information Sciences, Volume 174, Issues 3–4, 11 August 2005, Pages 219-235, ISSN 0020-0255, http://dx.doi.org/10.1016/j.ins.2004.10.005

[2 ]  Shen F, Hegglin MI, Luo Y, Yuan Y, Wang B, Flemming J, Wang J, Zhang Y, Chen M, Yang Q, Ge X. Disentangling drivers of air pollutant and health risk changes during the COVID-19 lockdown in China. npj climate and atmospheric science. 2022 Jun 30;5(1):54.

*Since the authors focus on the COVID-19 lockdown period, and how about the influence of emission reductions on the anomalies of PM2.5 and O3?*

**Authors' Reply #4:** We thank the reviewer for his/her thoughtful question. In general, anthropogenic reduction has dominated control of the decline in the primary air pollutants but the influence is complex on the secondary air pollutants, including $PM_{2.5}$ and $O_3$. For example, in NCP, our study here demonstrate that large $PM_{2.5}$ anomalies in SWP-I (-27.6 μg/m$^3$ ) and SWP-III (-27.9 μg/m$^3$) in 2020 were dominated by the anthropogenic reduction. However, the anomalies of $PM_{2.5}$ and $O_3$ in SWP-II are subject to regional variations  due to meteorology, which is what we are studying here specially in the haze event(***Revised Manuscript version Line****:299-306*).

*Overall, this paper was well organized, but I still find some explanations for lack of evidence. Please try to improve it.*

**Specific comments:**
*Lines 16 to 17: Please clarify the sentence.*

**Authors' Reply #5:** The reviewer might be confused by why North East China (NEC) was abruptly introduced. The reason is that North East China (NEC) and NCP were both significantly under the impact of the double-centre high-pressure system. We had stressed this unnecessary point. To avoid this confusion, NEC was deleted from this sentence(***Revised Manuscript version Line****:18)*.

*Line 75: What does the "100%" mean?*

**Authors' Reply #6:** We thank the reviewer for pointing out this inaccurate statement. This sentence should be "achieving the classification 100% automatically" (***Revised Manuscript version Line****:76)*.

*Lines 187 to 189: Please clarify the sentence.*

**Authors' Reply #7:** We thank the reviewer for his/her detailed reading of our previous article. We have now modified this ambiguous statement, which highlights the contribution of the meteorological effect(***Revised Manuscript version Line****:202-203)*.

*Lines 244 to 245: This sentence is unclear.*

**Authors' Reply #8:** We thank the reviewer for pointing out this confusing statement. We now have changed this sentence to "NCP and NEC (SC) have higher(lower) than expected concentrations for $PM_{2.5}$ ($O_3$)" (***Revised Manuscript version Line****:261-262)*.

---

## Author Comment (AC2)

**Reply Letter to the Comments**

**Manuscript ID:** Preprint egusphere-2023-2425
**Manuscript Title:** Impact of Weather Patterns and Meteorological Factors on $PM_{2.5}$ and $O_3$ during the Covid-19 Lockdown in China
**Authors:** Fuzhen Shen, Michaela I. Hegglin, Yue Yuan

Dear Editor, dear Reviewers,

We would like to thank you and the reviewers for the thoughtful comments regarding our manuscript. These comments have helped us greatly improve the interpretation of our findings, and are important for our future work as well. We have carefully revised the manuscript accordingly. Our point-to-point responses to the reviewers' comments, which are repeated in italics, are given below.

**To Reviewer #1**

*The authors adopt Self-Organizing Map (SOM) and Gradient Boosting Machine (GBM) to classify the weather patterns and identify key local meteorological factors contributing to the haze event characterized by the elevated PM2.5 levels in North China Plain and a slight decrease in ozone levels over Southern China (SC) from 21st January to 9th February 2020. The SOM classifies three types of synoptic-scale weather patterns during the first two months from 2015 to 2020 over China. Subsequently, the authors compare the changes in these three patterns in 2020 with the same period from 2015 to 2019, aiming to elucidate how synoptic patterns may facilitate the haze event in NCP and the decline in ozone levels over SC. On the other hand, the GBM reveals that temperature is the key local meteorological variable explaining the variations in PM2.5 over NCP and North East China (NEC). Surprisingly, it indicates that relative humidity (RH) plays a pivotal role facilitating ozone formation in SC, whereas temperature exerts a suppressing influence.*

*This work is well within the scope of ACP. The authors present sufficient results concerning the anomalies of PM2.5 and ozone during the study period. However, I'm skeptical about certain results of this study, particularly those derived from the GBM analyses. The acceptance of this manuscript is contingent upon the authors thoroughly validating the GBM results. In addition, several places in this manuscript require substantial improvement. I recommend the authors address the comments and concerns detailed below.*

**Authors' Reply #0:** Many thanks for the reviewer's positive summary of our manuscript and his/her helpful comments. We have followed the reviewer's suggestions and made modifications to the text accordingly. We hope to be able to convince you about the validity of our used SHAP approach with our more detailed answers to your concerns as follows below.

**General comment:**

*After reading this manuscript, my initial impression is that the writing should be improved to reduce ambiguity. For instance, certain lines (e.g., Lines 50 to 52) could be revised to clearly state that the ozone decline is a regional phenomenon, rather than ambiguously suggesting it as a nationwide phenomenon during the study period. Furthermore, the decline is very small (see Fig. 2d), and this point should be clearly emphasized in the main text. Besides, the abbreviation "SC" needs to be consistent, as I'm not sure what SC specifically refers to in this study—Southern China (Line 9 in the abstract)? Southwestern China (Line 31)? Or Southern Coast (line 109)? The description of GBM in section 2.3 should include more details, as it is unclear how the GBM is implemented. For example, how is the cross-validation conducted? What are the hyperparameters?*

**Authors' Reply #1:** Many thanks for the reviewer's suggestions. We have now changed the text in different locations to increase the clarity of our writing. For the ozone decline, we now clarify that it is such a regional and small decline in China (***Revised Manuscript version Line***:*203*). We introduce abbreviations at their first occurrence (e.g., SC for Southern Coast of China and SWC for Southwestern China, see ***revised Manuscript version Line***:*32* ).

We have now also included more information on the use of the GBM prediction model in section 2.3 (***Revised Manuscript version Line***:*167-174*). In particular, we explain more thoroughly that cross-validation is mainly used to estimate how accurately a predictive model will perform in practice[1]. To check the accuracy of the ML model used in our study, a time series split rolling cross-validation based on 5 splits was used, similarly to our earlier publication[2], for which data used for the training task always preceded the data used for validation. In detail, the ML training model was used for 2015, 2015-2016, 2015-2017, 2015-2018, and 2015-2019, while the testing of the model then was implemented over the 3 first months of 2016, 2017, 2018, 2019, and 2020, respectively[2]. The hyperparameters of the model that we used are as follows: 'number_leaves ', 'objective', 'min_data_in_leaf', 'learning_rate', 'feature_fraction', 'bagging_fraction', 'bagging_freq', and 'metric'(detailed parameter information of the model can be accessed from: https://lightgbm.readthedocs.io/en/latest/Parameters.html).

*Reference:*

[1] Cawley GC, Talbot NLC. On Over-fitting in Model Selection and Subsequent Selection Bias in Performance Evaluation. *J Mach Learn Res* 11, 2079–2107 (2010).

[2] Shen F, Hegglin MI, Luo Y, Yuan Y, Wang B, Flemming J, Wang J, Zhang Y, Chen M, Yang Q, Ge X. Disentangling drivers of air pollutant and health risk changes during the COVID-19 lockdown in China. npj climate and atmospheric science. 2022 Jun 30;5(1):54.

*In terms of the scientific aspect, I have two major concerns. First, the synoptic-scale weather patterns (SWPs) are classified and identified for the first two months of 2015 to 2020, whereas the haze event in NCP and ozone decline in SC occurred from 21st January to 9th February 2020. I'm afraid that the results of SWPs may be too broad to accurately interpret the study period (21st January to 9th February 2020). Second, the authors should validate the counterintuitive GBM results—RH facilitating and temperature suppressing ozone in SC. This does not align with the meteorological effects on ozone during summertime, as demonstrated by the studies of Li et al. (2019), Weng et al. (2022). I understand that there are differences in the study periods between this research and the other two; nonetheless, the fundamental mechanism by which meteorology influences ozone levels should be consistent. Arguably, the key meteorological factors may vary across seasons, but it is unlikely that temperature would suppress ozone formation. Therefore, I suggest that the authors undertake a more comprehensive discussion. For example, the authors could analyze the time series of temperature, RH together with ozone during their study period to investigate any unexpected correlations. It is also important to note that the impact of emission reductions may be equally important, particularly given the massive reductions during the study period.*

**Authors' Reply #2:** We thank the reviewer for his/her comments. Regarding your first concern, there may be a misunderstanding. We thus now seek to clarify the use of our methodological approach here and in the text. Firstly, we assess which weather patterns exist over China in a climatological sense. We then investigate which of these weather patterns are observed in 2020 and whether one of them is anomalously prevalent. To be able to evaluate this, we need to have more historical data than that only in 2020, otherwise we couldn't compare it to a climatological reference state. Only then we compare how these weather patterns influence the trace gas distributions and which meteorological factors were key drivers. On the other hand, to check whether our result is robust, we also ran the SOM model using 2020 data only (**Fig. S17**), and found that the three and seven weather pattern results are similar with our final results based on the longer time period. Overall, the above is the logic behind our model data selection.

Regarding your second concern, firstly, we agree with your point that temperature would not suppress ozone formation. From our result, you can see that both temperature and RH show a clear anti-correlation with the ozone during the haze event (Please see the modified ***Figure 11 and Figure S16***). However, it needs to be further clarified that the SHAP value of temperature for ozone prediction does not reflect the relationship between temperature and ozone directly, and it is also not the same as the Gini importance score in the article by *Weng et al.(2022)*[1]*,* which has no ability to determine the positive or negative effect on ozone prediction. As described in Method session 2.4, SHAP's local explanations can vary in being positive or negative, reflecting how predictors influence the predicted outcome[2]. This contribution is assessed from the base value (the predicted mean value) to the final model output. Variables that push the prediction to higher values are displayed in positive, while those pushing the prediction to lower values are shown in negative(***Revised Manuscript version Line****:183-187*). We now explain the negative SHAP value of temperature in ozone prediction from two perspectives.

From the perspective of the calculation method, as shown in ***Fig. 11(g)***, compared to the before haze event

in SC, relative smaller mean RH would be conducive to more ozone generation pushing the predicted ozone higher. In contrast, the smaller mean temperature during the haze event would suppress ozone production thus pushing the predicted ozone lower. As a result, SHAP values for temperature and RH are negative and positive respectively. From the perspective of physical mechanism, as shown in *Fig. 11(h)*, compared to the haze event from 27th Jan to 2nd Feb 2020 in SC, relative lower RH with greater absolute mean SHAP value (9.8) than temperature (-6.7) dominated the ozone increase, attributed to the SWP-II in SC (*Fig. 9(d, f)*) with strong moist winds from the ocean, leading to the importance of RH surpassing temperature, which is consistent with the study of *Weng et al.(2022)*. However, over the full time period of the haze event (21st Jan to 9th Feb), the negative effect of temperature dominated a higher ozone level during the haze event based on the larger absolute SHAP value for temperature (albeit, note that this has to be understood relative to a lower ozone level compared to climatology). This non-decreased response from ozone to lower temperature might be attributed to the uncoordinated emission reductions of ozone precursors (***Revised Manuscript version Line***:357-369).

[Figure]

*Reference:*

[1] Weng X, Forster G, Nowack P. A machine learning approach to quantify meteorological drivers of ozone pollution in China from 2015 to 2019. Atmospheric Chemistry and Physics. 2022 Jun 29;22(12):8385-402

[2] Lundberg, S.M., Erion, G., Chen, H. et al. From local explanations to global understanding with explainable AI for trees. Nat Mach Intell 2, 56–67 (2020). https://doi.org/10.1038/s42256-019-0138-9.

*The title of this manuscript should be revised. The phrase, "COVID-19 Lockdown" is too broad. This is because the authors focus on the period of haze event rather than the entire duration of COVID-19 Lockdown.*

**Authors' Reply #3:** We do not agree with the reviewer in this point. A haze event may influence the responses of pollutant concentrations under Covid-Lockdown differently than in other years, so it is important to keep this qualifier in the title. However, to find a compromise we now use a new title "Impact of Weather Patterns and Meteorological Factors on $PM_{2.5}$ and $O_3$ responses to the Covid-19 Lockdown in China" (***Revised Manuscript version Line:1-2***). This should express that anomalous weather patterns such as the haze event can have a specific impact, contrary to those found in other studies during the Covid-19 Lockdown generally assumed to result in lower $PM_{2.5}$ and higher $O_3$.

**Specific comments:**

*Line 14: What are the two scenarios? Please clarify.*

**Authors' Reply #4:** We acknowledge the reviewer's request for clarification on this point. Now the text in the abstract has been revised as: These scenarios implemented the comparisons of observation in 2020 with climatology averaged over the years 2015-2019 by a novel structural SOM (Self-Organising Map) model and with the prediction with 'Business As Usual' (hereafter referred to as BAU) emission strength by GBM (Gradient Boosting Machine) model, respectively(***Revised Manuscript version Line:14-17***).

*Line 16: I suggest specifying "Gradient Boosting Machine" instead of the generic term "ML (Machine learning)".*

**Authors' Reply #5:** As suggested, the generic term "ML (Machine learning)" has been replaced by "Gradient Boosting Machine" (***Revised Manuscript version Line:17, 156***).

*Lines 16 to 17: This sentence is unclear. Why North East China (NEC) is abruptly introduced?*

**Authors' Reply #6:** The reason is that North East China (NEC) and NCP were both significantly under the impact of the double-centre high-pressure system. We had stressed this unnecessary point. To avoid this confusion, NEC was deleted from this sentence(***Revised Manuscript version Line:18***).

*Lines 50 to 53: The later sentence, "In details, the substantial decrease…resulted in an increase in O3…" contradicts the "O3 downward trend" from the previous sentence*
.
**Authors' Reply #7:** We thank the reviewer for his/her detailed reading of our manuscript. To avoid this contradiction, the "O3 downward trend" has been deleted(***Revised Manuscript version Line:52***).

*Lines 70 to 71: This statement is too generic. Air quality is not necessarily improved under cyclonic weather systems. It is dependent on the types and positions of the cyclonic weather systems. For example, see Jiang et al. (2015), Wang et al. (2022).*

**Authors' Reply #8:** We thank the reviewer for this helpful comment. As suggested by the reviewer, we have modified this sentence to "good air quality conditions are often observed under cyclonic weather systems with certain types and positions" and added the suggested references(***Revised Manuscript version Line:71***).

*Line 75: what does "100%" refer to here?*

**Authors' Reply #9:** We thank the reviewer for pointing out this inaccurate statement. This sentence should be "achieving the classification 100% automatically". Please see the amended text(***Revised Manuscript version Line***:76).

*Line 155: Yes, predictions using conventional CTMs rely on accurate emission inventories. Besides, uncertainties can also be derived from the chemical mechanism (e.g., Knote et al., 2015; Weng et al., 2023).*

**Authors' Reply #10:** We thank the reviewer for supporting our point and providing a more comprehensive statement. I have added this sentence and relative references to the main text(***Revised Manuscript version Line***:160-161).

*Line 175: Please provide more detailed explanations regarding equation (6).*

**Authors' Reply #11:** We acknowledge the reviewer's request for clarification on this point. As applied in equation (6), SHAP's local explanations can vary in being positive or negative, reflecting how predictors influence the predicted outcome. In contrast, other ML methods typically yield a single positive value, indicating overall importance. Specifically, in local interpretability analysis, SHAP indicates the contribution of each variable to the prediction of a specific sample. This contribution is assessed from the base value (the predicted mean value) to the final model output. Variables that decrease the prediction are displayed in negative, while those increasing the prediction are shown in positive(***Revised Manuscript version Line***:185-193).

*Lines 187 to 189: This sentence is ambiguous. If I understand the authors' previous work correctly, the -0.9% decline of ozone is driven by the meteorological effect (Shen et al., 2022), not an observed -0.9% decline.*

**Authors' Reply #12:** We thank the reviewer for his/her detailed reading of our previous article. We have now modified this ambiguous statement, which highlights the contribution of the meteorological effect(***Revised Manuscript version Line***:203-205).

*Line 200: Title of table 1 should be revised. I don't think these are the mean averages.*

**Authors' Reply #13:** We thank the reviewer for catching this mistake, which we now have corrected accordingly(***Revised Manuscript version Line***:220).

*Line 208: What does "climatology" refer to here?*

**Authors' Reply #14:** We thank the reviewer for pointing out this incomplete statement. The complete statement should be "climatology averaged over the years 2015-2019" (***Revised Manuscript version Line***:225).

*Lines 244 to 245: Similar to lines 16 to 17, I find this sentence confusing, as I'm not sure which of NCP, NEC and SC has higher/lower PM2.5/ozone.*

**Authors' Reply #15:** We thank the reviewer for pointing out this confusing statement. We now have changed this sentence to "NCP and NEC (SC) have higher(lower) than expected concentrations for PM2.5(O3)"

(***Revised Manuscript version Line***:261-262).

*Line 247: what are "the days" here? The first two months? Same calendar days of the haze event? Similar issues can be found in the captions of Figure 5 to 10.*

**Authors' Reply #16:** We thank the reviewer for pointing out this unclear statement, which we clarified in the captions of Figures 5 to 10 (***Revised Manuscript version Line***:245, 280, 284, 289, 294, 313). The first two months in 2020 refer to January and February 2020 (***Revised Manuscript version Line***:104).

*Line 292: "... is leading to a more moderate ozone increases ..." I suggest avoiding the term "leading to" here as it implies ozone increases are solely driven by meteorology. In addition, as mentioned in the general comment, I suggest giving detailed discussion regarding why negative temperature anomaly can contribute to ozone increases.*

**Authors' Reply #17:** Thank you for this suggestion. This change has been implemented accordingly, see ***Line 309***, where the sentence now reads: '*... is favouring to a moderate ozone increase ...* '. Meanwhile, as suggested by the reviewer, we have added a detailed discussion to the main text. The statement now reads: In terms of the positive SHAP values of temperature, pressure, RH, and WS in PM2.5 predictions, it reveals that those meteorological features push PM2.5 prediction to a higher value, suggesting the final predictions were up to the baseline concentrations in NCP and NEC. In SC (Fig. 11(i)), positive mean SHAP values (2.2 $\mu g/m^3$) would be conducive to additional ozone generation due to the relative lower RH compared to the before haze event (Fig. 11(g)), thus pushing the predicted ozone higher. In contrast, negative mean SHAP value (5.5 $\mu g/m^3$) during the haze event (Fig. 11(i)) would suppress ozone production attributable to relative smaller mean temperature thus ledading to a lower ozone prediction. It should be noted that RH with higher SHAP value (9.8 $\mu g/m^3$) overpassing temperature (-6.7 $\mu g/m^3$) become the primary factor dominating the high ozone level from 27th Jan to 2nd Feb 2020. It is attributed to the SWP-II in SC (Fig. 9(d, f)) with strong moist winds from the ocean, leading to the importance of RH over temperature, which is consitent with the previous study(Weng et al., 2022). However, over the full period of the haze event, the negative effect of temperature dominated a higher ozone level during the haze event based on the larger absolute SHAP value for temperature. This non-decreased ozone response to lower temperature might be attributed to the uncoordinated emission reductions of ozone precursors. (***Revised Manuscript version Line***:357-369).

*Line 321: It is unclear to me how this "business as usual" scenario is constructed. Please elaborate.*

**Authors' Reply #18:** We appreciate the reviewer's comment. We now clarify in the text (***Revised Manuscript version Line***:339-341) that the "*business as usual*" scenario is constructed by the Gradient Boosting Machine that trained a model using historical weather factors and ozone. The GBM model can then be used based on prevailing meteorology to predict ozone as if the huge emission reduction due to the COVID-19 lockdown would not have happened. It is a counterfactual scenario assuming the emission strength is the same, thus being called "business as usual".

*Line 341: Following my general comment above, could the higher wind speed be the predominant factor causing the decline in ozone in SC? If this is the case, why is this not depicted in Figure 11 (h)?*

**Authors' Reply #19:** We thank the reviewer for his/her thoughtful point. Through our responses to your general comment above, I believe you now understand what is the difference between the relationships between the SHAP of temperature versus predicted ozone and observational temperature versus observational ozone. You would acknowledge our result presentation is nevertheless valid. As a result, temperature is regarded as the predominant factor due to a higher absolute SHAP value (-5.5 µg/m$^3$) than that for RH (2.2 µg/m$^3$) in SC. It should be clarified that RH was wrongly regarded as the primary driver due to the overestimation of its SHAP value when we examined the data.